# On the Pressure and Rate of Infiltration Made by a Carbon Fiber Yarn with an Aluminum Melt during Ultrasonic Treatment

**Sergei Galyshev *** [ID]**, Bulat Atanov and Valery Orlov**

Osipyan Institute of Solid State Physics RAS, 142432 Chernogolovka, Russia
* Correspondence: galyshew@gmail.com

**Abstract:** The effect of the infiltration time of a carbon fiber yarn in the range of 6 to 13.6 s on the infiltrated volume under the cavitation of an aluminum melt has been studied. When the infiltration time was more than 10 s, the carbon fiber was completely infiltrated with the matrix melt, and a decrease in the infiltration time led to a monotonous decrease in the fraction of the infiltrated volume. Based on the experimental data, the infiltration rate and the pressure necessary to infiltrate a carbon fiber yarn with an aluminum melt were estimated. The infiltration rate was 20.9 cm$^3$/s and was independent of the infiltration depth. The calculated pressure necessary for the complete infiltration of a carbon fiber yarn at this rate was about 270 Pa. A comparison of the pressure values calculated according to Darcy's and Forchheimer's laws showed that the difference between them did not exceed 0.01%. This indicates that a simpler Darcy's law could be used to estimate pressure.

**Keywords:** aluminum matrix; carbon fiber; infiltration; infiltration rate; infiltration pressure





## 1. Introduction

The development of the technology for producing a carbon–aluminum composite dates back to the late 1960s (the group of A. Kelly, England, and A.A. Khvostunkov in the group of S.T. Mileiko, Institute of Solid State Physics, USSR Academy of Sciences [1]). The main problems of producing aluminum carbon are as follows: first, the non-wetting of the carbon fiber by the aluminum melt, and second, the chemical interaction between the matrix and the fiber with the formation of aluminum carbide Al4C3. The formation of aluminum carbide is highly undesirable for several reasons. Firstly, Al4C3 is highly hygroscopic, which can cause corrosion. Secondly, the chemical interaction introduces coarse defects on the surface of the fiber, which significantly reduces its effective strength, and, hence, the strength of the carbon fiber/aluminum matrix composite [2]. Thirdly, this interaction results in the formation of a "strong" matrix/fiber interface, which is undesirable in most cases due to the need for the uniform distribution of fibers in the matrix and the limitation of the fiber volume fraction (not higher than 30%) [1].

To solve the first problem, the most common method for producing composites with an aluminum matrix and carbon fiber is the one-stage liquid-phase infiltration of the matrix melt under the action of external pressure [3,4]. However, this method has several drawbacks, the most significant of which is the deterioration of the mechanical properties of a composite as a result of the chemical interaction between the matrix and the fiber due to the long time required for infiltration. In other words, a solution to the first problem creates a second problem.

Since the first carbon fiber/aluminum matrix composite was obtained, many attempts have been made to solve this second problem, namely, including the formation of aluminum carbide at the matrix/fiber interface. One of them is to create a barrier layer at the matrix/fiber interface. Such coatings, in addition to limiting the chemical interaction, can play the role of "weak boundaries", which is very attractive. However, as a rule, an expensive and low-performance chemical vapor deposition method is used to create these coatings [5].

Another solution is to reduce the time of contact between the fiber and the melt. This method is most clearly presented in [6], where a carbon fiber/aluminum matrix wire was obtained by pulling the carbon fiber through a pure aluminum melt at 730 °C. The authors showed that the amount of carbide directly depended on the time of contact between the fiber and the matrix melt, while with an increase in the carbide phase content, the strength of the composite monotonically decreased.

This solution is more technologically advanced, and its practical application is the two-stage method. The first stage is to produce a carbon-aluminum wire by drawing a carbon fiber yarn through a matrix melt that has been subjected to ultrasonic treatment [7,8]. In the second stage, a preform is compacted from the wire by hot pressing. The advantages of this method are the ability to significantly reduce the time of contact between the matrix melt and the fiber in the first stage [8] and considerably reduce the temperature of obtaining bulk material in the second stage. This significantly reduces the chemical interaction between the matrix and the fiber, ensuring high mechanical properties in the composite. Another advantage of the two-stage method is manifested when large-sized products need to be created from a carbon–aluminum composite. The two-stage method does not require powerful technological equipment, such as hydraulic presses or gas presses, which are necessary to create high external pressure in the case of the one-stage liquid-phase infiltration.

The most important parameters in both of the mentioned methods are the infiltration pressure and rate since it is thanks to them that carbon fiber is introduced into the aluminum matrix, and it is these parameters that determine the measure of the chemical interaction between the matrix and the fiber and, ultimately, the mechanical properties of a composite. In addition, the numerical expression of the pressure dependence of the infiltration rate is mandatory for constructing the process of producing such composites.

When utilizing the method of one-stage liquid-phase infiltration under the action of external pressure, this parameter is usually estimated using the thermodynamic and hydrodynamic approaches [9–11]. Pressure estimation using the thermodynamic approach is based on the equality of the work performed during infiltration and the energy of the formation of a new surface between the matrix melt and the fiber [9]; it can be expressed as follows:

$$P \cdot \Delta V = \sigma \cdot S, \tag{1}$$

where $P$ is the pressure due to which th3 system volume changes $\Delta V$, and $\sigma$ and $S$ are the specific surface energy and the area of a new surface between the matrix melt and the fiber, respectively.

It should be noted that this approach allows not only the pressure to be estimated at which the liquid and the fiber are in equilibrium. In other words, this pressure is necessary to compensate for the Laplace capillary pressure but is not a sufficient condition for the [9–12] infiltration process. Moreover, this approach does not enable the estimation of the infiltration rate.

To estimate the infiltration rate depending on the pressure, the hydrodynamic approach can be used based on Darcy's law:

$$\frac{P}{L} = \frac{\eta}{k} \cdot \frac{Q}{F}, \tag{2}$$

where $k$ is the permeability of the porous medium, $\eta$ is the dynamic viscosity of the liquid, $P$ is the pressure drop in the direction of the liquid flow, $F$ is the cross-sectional area of the porous medium, $L$ is the length of the porous medium, and $Q$ is the fluid flow that is filtered through the porous medium.

The application of this law agrees well with the data of experiments on the infiltration of fiber with a matrix melt under the action of constant external pressure [12]. Note that since this hydrodynamic approach does not allow for Laplace capillary pressure, the infiltration pressure can be expressed as the sum of pressures calculated using the described approaches.

However, the question remains as to whether Darcy's law can be used to estimate the infiltration pressure when producing a composite wire by drawing a carbon fiber yarn through the melt of a matrix that has been subjected to ultrasonic treatment. This question exists since, in this case, the pressure is created due to cavitation. At the moment of collapse for the cavitation bubbles, microflows arise, the local pressure of which can reach several GPa. This results in an increase in the pressure in the cavitation area. In this regard, it can be assumed that, in this case, Forchheimer's [13] law is more suitable for estimating the rate and pressure of infiltration; in the scalar form, this can be expressed as follows:

$$\frac{P}{L} = \frac{\eta}{k} \cdot \frac{Q}{F} + \frac{\rho \cdot \beta}{\sqrt{k}} \cdot \left(\frac{Q}{F}\right)^2, \tag{3}$$

where $\rho$ is the density of the liquid, and $\beta$ is the Ergun constant.

This work aimed to answer the above questions and estimate the pressure and rate of the infiltration of carbon fiber under the action of melt cavitation based on experimental data.

## 2. Materials and Methods

As a reinforcement, a commercially available continuous carbon fiber yarn of the UMT40-3K-EP brand (manufactured by UMATEX Group, Russia) was used; it was thermally cleaned from sizing in a vacuum at 400 °C for 15 min. Alloys of the Al-Sn system were used as the matrix material. The tin content was 25 at.%.

### 2.1. Production of a Composite Carbon-Aluminum Wire

The scheme for producing a carbon-aluminum wire was similar to that described in [7,8] (Figure 1).

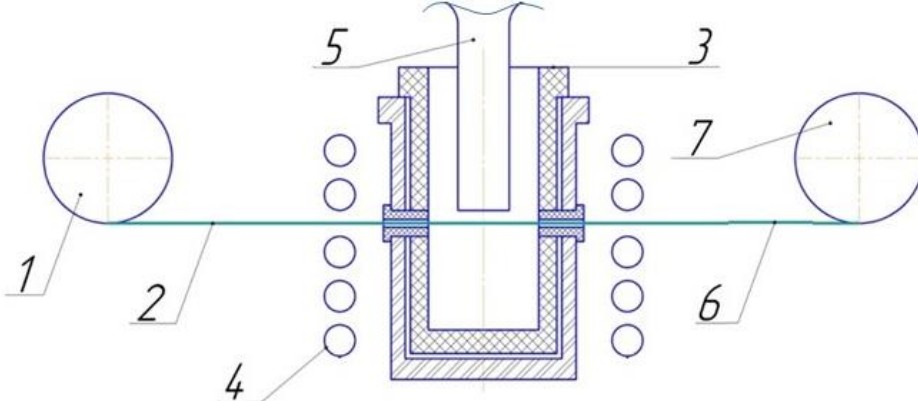

**Figure 1.** Scheme for producing a carbon-aluminum wire.

From the output coil (1), carbon fiber yarn (2) entered the matrix melt inside the crucible through the inlet die in the wall of the graphite crucible (3). The crucible was heated by induction heating (4). To ensure infiltration in the matrix melt, cavitation was created using a niobium waveguide (5). After passing through the outlet die in the crucible wall, the fiber infiltrated with the melt and entered the cold region. After the crystallization of the melt, the composite wire was produced in this way and wound onto a receiving spool (7). The mechanical properties of the composite wire obtained by this method and their comparison with the properties of similar composites are described in [8].

The time of residence of the carbon fiber in the melt was controlled by adjusting the rotation speed of a stepper motor with a receiving coil attached to its shaft. The time control accuracy was at least 0.1 s. The melt temperature was controlled by a thermostat. The amplitude of fluctuations in the melt temperature did not exceed 3 °C. The control thermocouple was located directly in the matrix melt. The ultrasonic treatment of the melt was carried out using a LUZD-1.5/1 system (manufactured by Kriamid LLC, Moscow,

Russia). The power of the ultrasonic treatment was constant at 1.5 W/mm$^2$. This was selected empirically in preliminary experiments.

The effect of the drawing rate on the infiltration of a carbon fiber yarn with an aluminum melt was studied at 600 °C in the speed range of 90 to 200 mm/min. Since the fiber path in the cavitation region was 20 mm, integer speed values gave fractional time values. For example, at speeds of 200 and 90 mm/min, it took about 6 and 13.6 s to pass through the cavitation area, respectively.

### 2.2. Scanning Electron Microscopy

The wire microstructure was investigated using a SUPRA 50VP high-resolution scanning electron microscope. The images were obtained in the secondary electron mode with an accelerating voltage of 5 kV at magnifications of up to 50,000×. The volume fraction of the fiber in the matrix and that of the non-infiltrated part in the wire were determined by the point method for at least three different cross-sections of the composite wire at a magnification of 500×, i.e., in almost the entire cross-sectional area of the wire.

The point method consisted of imposing an orthogonal grid on the SEM image of the wire cross-section. Next, the grid nodes that fell on the non-infiltrated area were counted. Their ratio to the total number of grid nodes that fell on the cross-section determined the volume fraction of the non-infiltrated volume. The distance between the neighboring grid nodes in terms of the scale of the SEM image was 10 μm, which made it possible to determine the presence of at least half of them as large pores.

## 3. Results

### 3.1. Effect of Time on the Infiltration of Carbon Fiber with a Matrix Melt

Figure 2 shows photos of the microstructure of the transverse sections of the wire produced with an infiltration time of 6 to 10 s. When the time was above 10 s, the carbon fiber yarn was completely infiltrated with the matrix melt. Since the samples obtained at 11.4 and 13.6 s were similar to a fully infiltrated sample at 10 s, their structure is not shown here. In addition, it is logical to assume that a further increase in time would also ensure complete infiltration. At 8.6 s, small non-infiltrated fiber clusters were observed in the bulk of the composite wire (dark areas in the images). Reducing the time to 8 s led to an increase in the number of non-infiltrated clusters. At 7 s, one large cluster of the non-infiltrated fiber and a lot of small ones were observed in the bulk of the composite wire. With a further decrease in time, the size of the large non-infiltrated cluster gradually increased. Regardless of the time, the volume fraction of the fiber in the infiltrated part of the yarn was almost unaffected and amounted to 43.6 ± 2.9%.

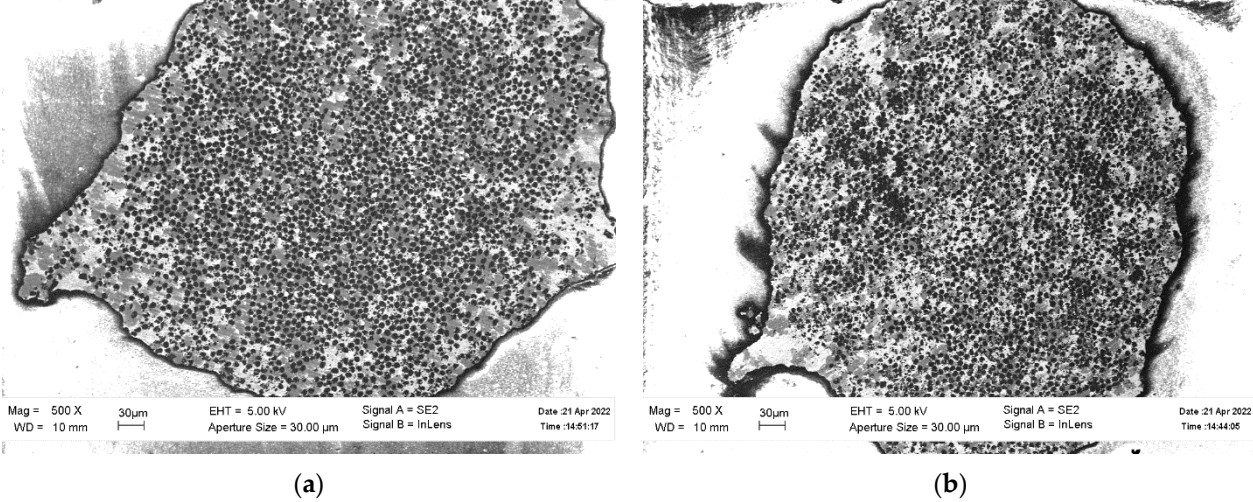

(a)　　　　　　　　　　　　　　　　(b)

**Figure 2.** *Cont.*

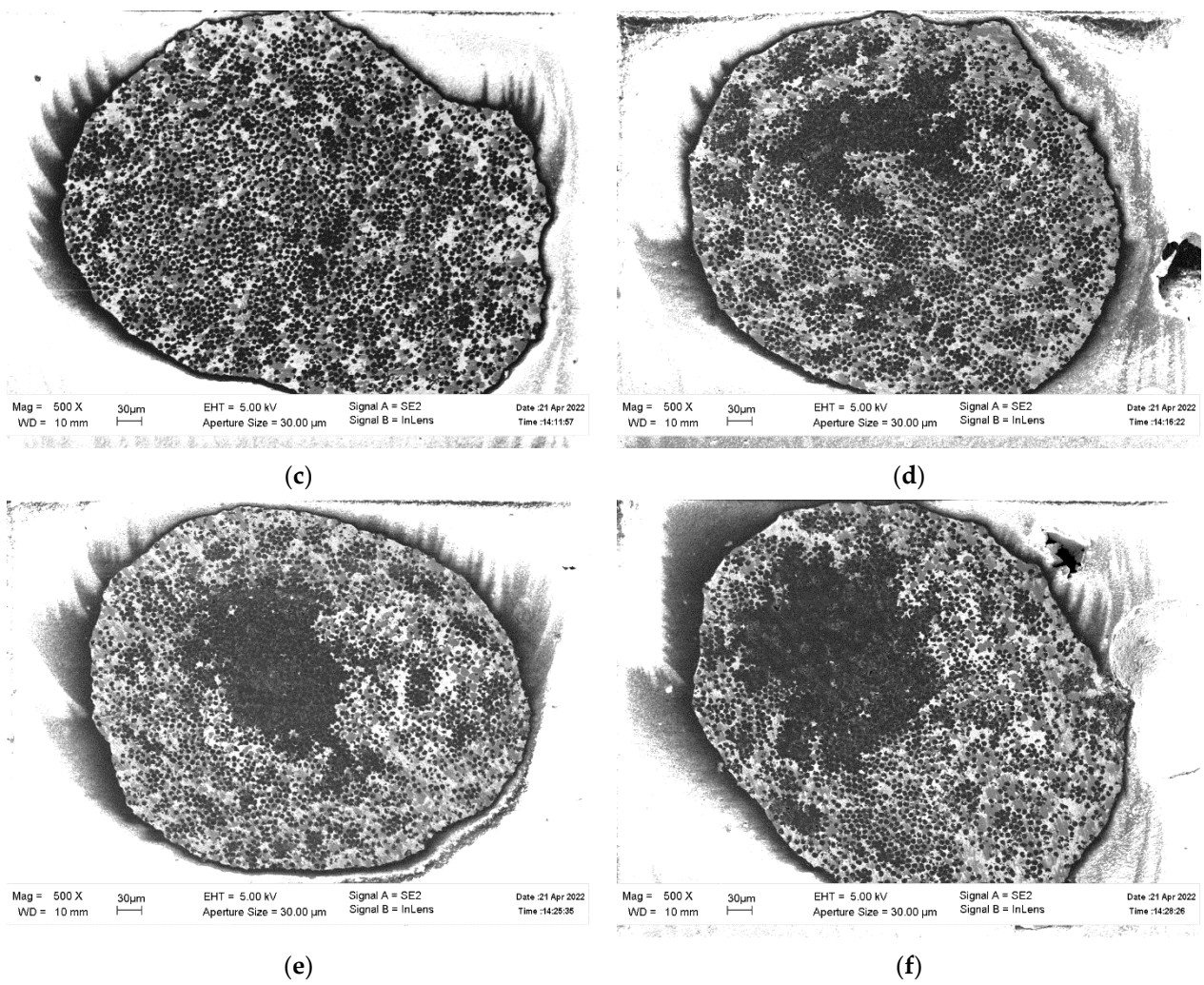

**Figure 2.** Microstructure of the transverse sections of the composite wire obtained at (**a**) 10 s, (**b**) 8.6 s, (**c**) 8 s, (**d**) 7 s, (**e**) 6.6 s, and (**f**) 6 s.

The dependence of the fraction of the infiltrated volume of the composite wire on the infiltration time is illustrated in Figure 3. The resulting dependence was close to the linear one, and the ratio of the infiltrated volume to the infiltration time was $20.9 \pm 1.54 \text{ cm}^3/\text{s}$. In other words, the infiltration rate was almost constant.

### 3.2. Estimation of Infiltration Pressure

A thermodynamic assessment of the pressure was made based on the equality of the work conducted during infiltration and the energy of the formation of a new surface between the matrix melt and the fiber, Equation (1). After simple mathematical transformations, Equation (1) could be written as [9]:

$$P = \frac{4V_f \cdot \sigma_{LA} \cdot \cos\theta}{d \cdot \left(1 - V_f\right)}, \tag{4}$$

where $P$ is the pressure, $V_f$ is the fiber volume fraction, $d$ is the diameter of one filament, $\sigma_{LA}$ is the specific energy of the liquid–atmosphere interface, and $\theta$ is the angle of the wetting of the fiber surface with the matrix melt.

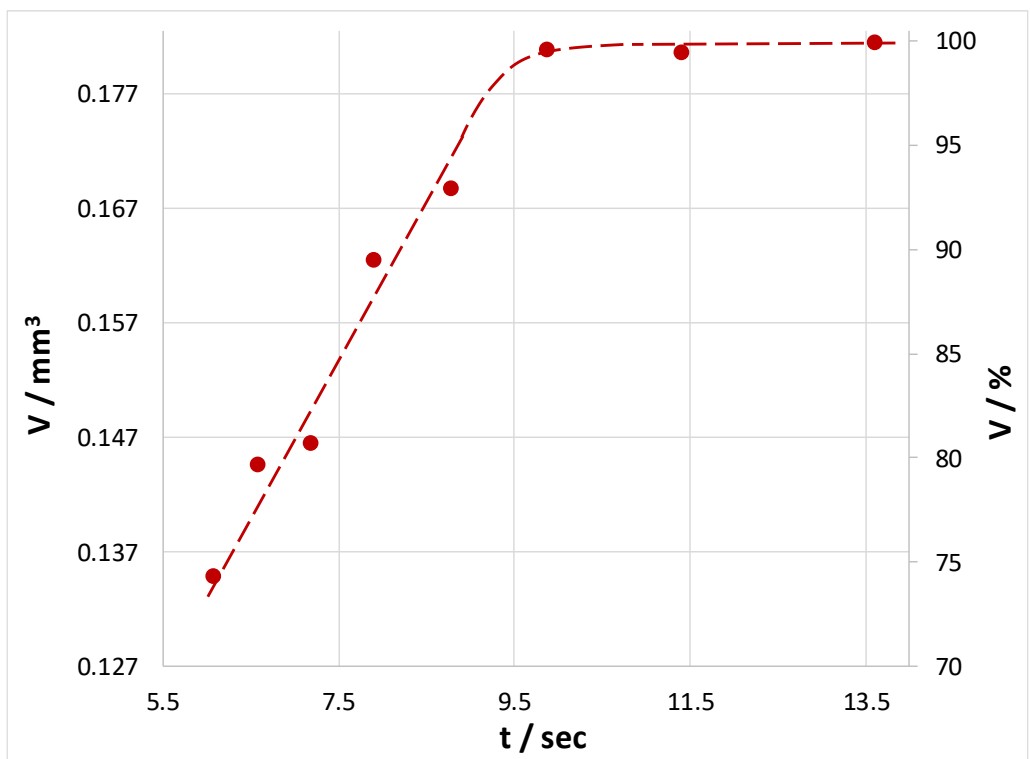

**Figure 3.** Dependence of the fraction of the infiltrated volume of the composite wire on the infiltration time.

Using the data from [14–17] and expression (2), the pressure required for infiltration was determined according to the thermodynamic estimate; this was $207 \pm 13$ Pa.

For the hydrodynamic assessment of the infiltration pressure of a carbon fiber yarn of unit length, a simplified model was considered, which had the following assumptions:

The carbon fiber yarn is a homogeneous porous medium.

The infiltration front has a cross-sectional shape that is close to a circle.

The infiltration front propagates evenly and only in directions perpendicular to the fiber axis.

A schematic representation of this model is shown in Figure 4a. Here, the arrows indicate the direction of the liquid flow, the infiltrated part of the porous medium of the length $L$ is highlighted in blue, $R$ is the radius of the infiltrated sample, and $F$ is the area of the infiltration front. Figure 4b illustrates an equivalent model, which is the infiltration of a porous medium on a variable cross-section.

In this case, the infiltration front area $F$ depended on the length of the infiltrated region $L$, namely:

$$F = 2\pi\cdot(R - L), \tag{5}$$

The pressure was estimated based on Darcy's law (Equation (2)) for low infiltration rates and Forchheimer's law (Equation (3)) for high rates. Since during the infiltration of a carbon fiber yarn with liquid aluminum, the cross-section of the porous medium was not constant and depended on the length of the infiltrated area (Equation (5)), the infiltration pressure for Darcy's and Forchheimer's laws, respectively, could be expressed as follows:

$$P = \int_0^L \left( \frac{\eta\cdot Q}{k\cdot 2\pi\cdot(R - L)} \right) dL, \tag{6}$$

$$P = \int_0^L \left( \frac{\eta\cdot Q}{k\cdot 2\pi\cdot(R - L)} + \frac{\rho\cdot\beta\cdot Q^2}{\sqrt{k}\cdot 4\pi^2\cdot(R - L)^2} \right) dL, \tag{7}$$

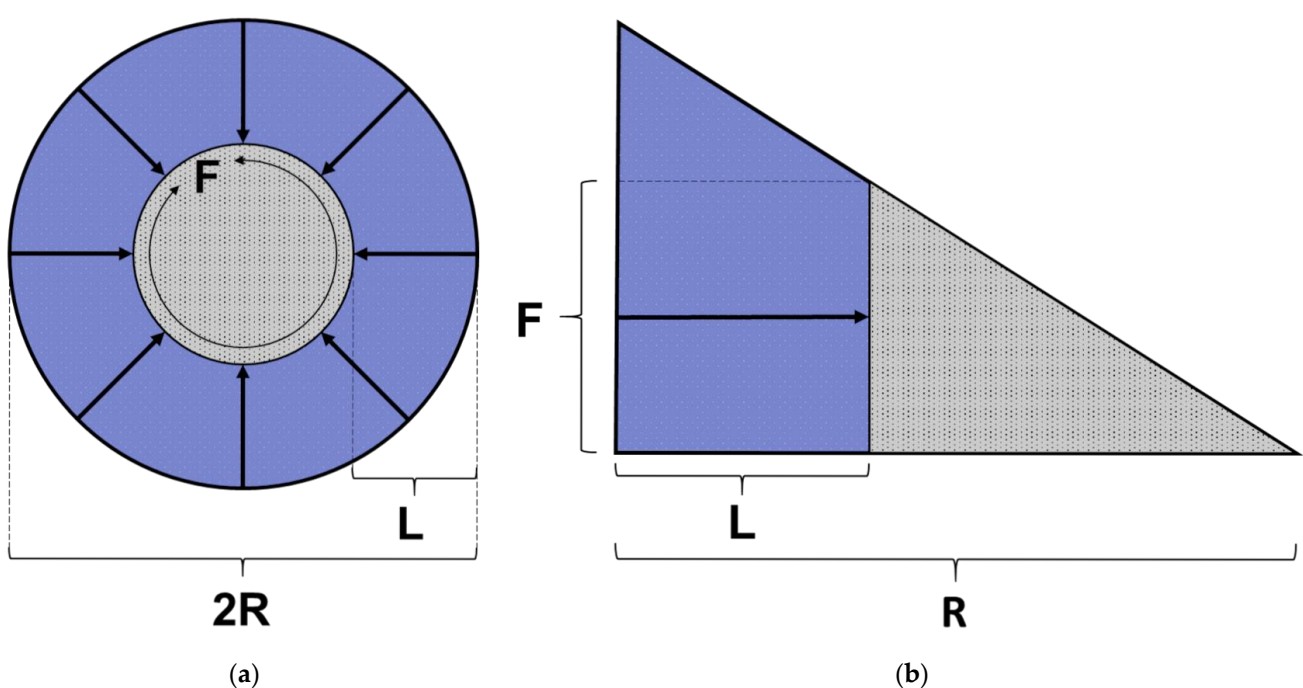

**Figure 4.** Schematic representations of the model of carbon fiber yarn infiltration (**a**) and an equivalent process (**b**).

Taking into account that the ratio of the infiltrated volume to its infiltration time was almost constant (Figure 3), i.e., that $Q$ was constant, after the integration, Equations (6) and (7) take the form:

$$P = \frac{\eta \cdot Q}{2\pi \cdot k} \cdot \ln\left(\frac{R}{R-L}\right), \tag{8}$$

$$P = \frac{\eta \cdot Q}{k \cdot 2\pi} \cdot \ln\left(\frac{R}{R-L}\right) + \frac{\rho \cdot \beta \cdot Q^2}{\sqrt{k} \cdot 4\pi^2} \cdot \left(\frac{1}{R-L} - \frac{1}{R}\right), \tag{9}$$

Thus, ceteris paribus, the infiltration pressure $P$, depends only on the length of the infiltrated porous medium $L$. Figure 5 shows the dependence of the pressure necessary for the infiltration of a carbon fiber yarn with the matrix melts on the path length of the infiltration front.

These dependencies were obtained based on the experimental data and Equations (4) and (8). The straight line "$P_{TD}$" in the figure shows the pressure calculated according to thermodynamics (Equation (4)). The gray dotted line indicates the error limits. The curved line "$P_{TD} + P_{GD}$" is the dependence of the pressure on the path length of the infiltration front, which was calculated based on the experimental data. The pressure value at each point is the sum of the pressures calculated using the thermodynamic and hydrodynamic approaches.

Note that the pressure necessary for the complete infiltration of a carbon fiber yarn at a rate of 20.9 cm$^3$/s was calculated in this way. It was 267.28 Pa in the case of using Darcy's law (Equation (8)) and 267.29 Pa in the case of using Forchheimer's law (Equation (9)). At the same time, the permeability of the porous medium calculated according to the data of [10] was 2.89*10$^{-13}$ m$^2$, the dynamic viscosity and density of the melt were taken to be 1.5*10$^{-3}$ Pa*s and 4300 kg/m$^3$, according to the data of [18–22], the Ergun coefficient calculated according to [23] was 0.17.

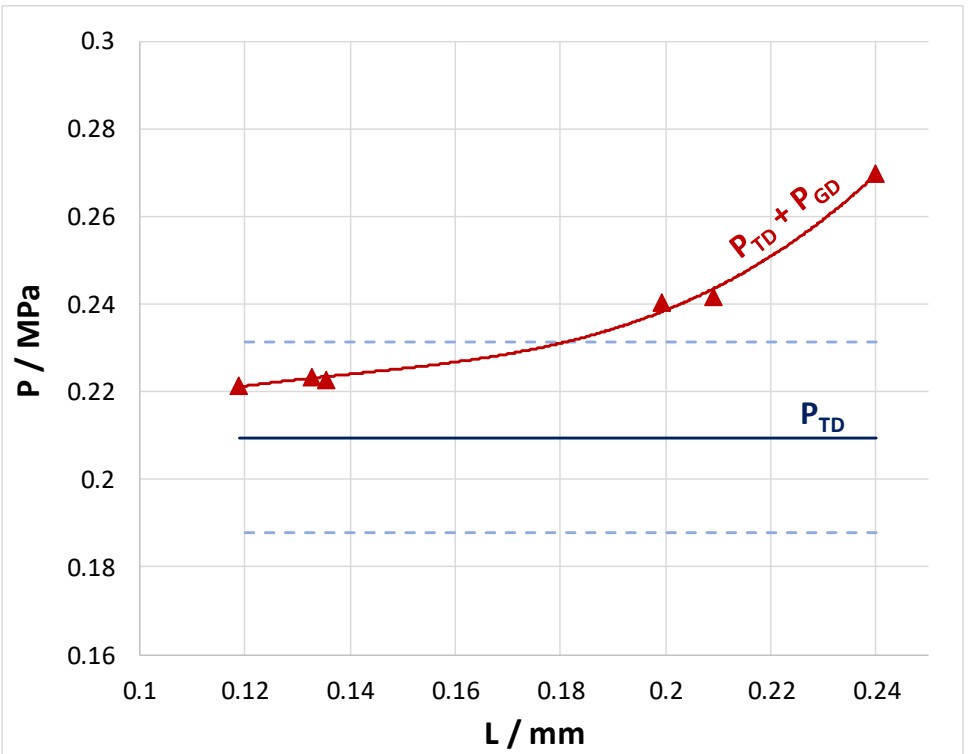

**Figure 5.** Dependence of the infiltration pressure on the length of the infiltrated porous medium.

## 4. Discussion

When the infiltration time was above 10 s, the carbon fiber was completely infiltrated with the matrix melt; a decrease in the time below this value led to a monotonous decrease in the infiltrated volume fraction. A similar phenomenon was observed in [14,15] and could be related to the infiltration rate, i.e., at a time below 10 s, the melt did not have time to infiltrate the entire carbon fiber yarn. At the same time, the data obtained show that the infiltration rate was almost constant. If we assume that the pressure created by cavitation in the melt had some average, constant value at the macro level, the constancy of the rate would not have agreed with Equations (2), (3), (8) and (9). Namely, at the constant external cavitation pressure $P$, the dynamic viscosity of the melt $\eta$, and the permeability of the porous medium $k$, causing an increase in the length of the infiltrated porous medium $L$ and a decrease in the cross-section of the infiltration front $F$, which took place during the infiltration process, made the equality impossible.

In [15], the authors observed a change in the size of a carbon fiber yarn in a cavitation field, which indicated a change in the distance between the individual filaments. This meant that a possible increase may have taken place in the permeability of the porous medium under the action of a cavitation field. In addition, since cavitation bubbles are formed more easily at the interface [24], the cavitation effect may increase as the carbon fiber is infiltrated due to an increase in the number of nucleation centers at the fiber–melt interface. An increase in the number of cavitation bubbles, in turn, may lead to a local increase in pressure. Moreover, it can be assumed that the dynamic viscosity of a liquid in a state of conditional rest and in a state of cavitation may be different.

Thus, the experimentally established constancy of the infiltration rate indicates a change in at least one of the following parameters during the infiltration process: pressure, the permeability of the porous medium or dynamic viscosity. Hence, it follows that classical infiltration laws do not fully reflect the mechanisms of the process under consideration. However, they can be used for approximate estimates of the infiltration pressure. This is due to the fact that the pressure values obtained with their help were overestimated since

they did not allow for a possible change in the medium permeability and the liquid viscosity. This overpressure, in turn, ensured that the fiber was infiltrated with the matrix melt.

A comparison of the pressure values calculated according to Darcy's and Forchheimer's laws has demonstrated that the difference between them did not exceed 0.01%. This indicates that, within the conditions under consideration, the contribution of the nonlinear component was extremely small. This means that a simpler Darcy's law could be used to estimate pressure.

**5. Conclusions**

1.  The effect of the infiltration time of a carbon fiber yarn in the range of 6 to 13.6 s on the infiltrated volume under the cavitation of an aluminum melt was studied. When the infiltration time was more than 10 s, the carbon fiber was completely infiltrated with the matrix melt, and a decrease in the infiltration time led to a monotonous decrease in the fraction of the infiltrated volume.
2.  Based on the experimental data, the infiltration rate and the pressure necessary to infiltrate a carbon fiber yarn with an aluminum melt were estimated. The infiltration rate was 20.9 cm$^3$/s and was independent of the infiltration depth. The calculated pressure necessary for the complete infiltration of the carbon fiber yarn at this rate was about 270 Pa.
3.  A comparison of the pressure values calculated according to Darcy's and Forchheimer's laws showed that the difference between them did not exceed 0.01%. This indicates that a simpler Darcy's law could be used to estimate pressure.

**Author Contributions:** Conceptualization, S.G. and V.O.; methodology, S.G.; validation, S.G.; formal analysis, B.A.; investigation, B.A.; resources, S.G.; data curation, S.G.; writing—original draft preparation, S.G.; writing—review and editing, S.G.; visualization, S.G.; supervision, S.G.; project administration, S.G.; funding acquisition, S.G. All authors have read and agreed to the published version of the manuscript.

**Funding:** This research was funded by the Russian Science Foundation, grant number 22-79-10064.

**Institutional Review Board Statement:** Not applicable.

**Informed Consent Statement:** Not applicable.

**Data Availability Statement:** Not applicable.

**Acknowledgments:** The authors are grateful to Evgeniya Yurevna Postnova and Elena Yurevna Aksenova. The research part of the work was carried out at the Collective Use Center of ISSP RAS. The corresponding author is grateful for inspiration to Yuri Burlan and the System Vector Psychology team.

**Conflicts of Interest:** The authors declare no conflict of interest.

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
