# Peer review of "On the Pressure and Rate of Infiltration Made by a Carbon Fiber Yarn with an Aluminum Melt during Ultrasonic Treatment"

_fibers, doi:10.3390/fib11050041_

Round 1
Reviewer 1 Report
Dear Author,
The manuscript entitled "On the Pressure and Rate of Infiltration of a Carbon Fiber Yarn with an Aluminum Melt During Ultrasonic Treatment” studied “The effect of the infiltration time of a carbon fiber yarn in the range of 6 to 13.6 seconds”. It is a good study, but some points need to be improved in terms of journal quality.
It does not make sense for the authors to mention that the study focused between 6 and 13.6 seconds since for values above 10 seconds, no results are shown.
- Line 45: What does the acronym "CVD" mean? You must write the term you want to abbreviate with the abbreviation. And since you don't appear any more throughout the text, the abbreviation doesn't even make sense;
- Line 66/67: “aluminum” error, should be replaced by aluminium;
- Session “2. Materials and Methods” It would be interesting to let the reader know the relevant (mechanical and physical) properties and to have a comparison in terms of future work by other authors.
- Figure 2, why is the study at 6, 6.6, 7, 8, 8.6 and 10 seconds? What is the logic? Why not study at 7.6 and 10.6 seconds, for example?
- Figure 3: If the experimental results were obtained between 6 and 10 seconds (for example, figure 2), how do the authors present us with a trend line for values below 6 and above 10 seconds? And how do they justify that for values greater than 10 seconds, the trend is completely different (constant)? At least one value greater than 10 seconds must be added to the graph to justify this result. The same for values less than 6 values, to justify the trend.
- Line 192: Figure 5 does not exist.
The English must be revised and improved.
Author Response
- It does not make sense for the authors to mention that the study focused between 6 and 13.6 seconds since for values above 10 seconds, no results are shown.
Infiltration results at 11.4 and 13.6 seconds are added to Figure 3. (page 6 line 166)
- Line 45: What does the acronym "CVD" mean? You must write the term you want to abbreviate with the abbreviation. And since you don't appear any more throughout the text, the abbreviation doesn't even make sense;
CVD stands for chemical vapour deposition. Text has been changed. (page 1 line 8)
- Line 66/67: “aluminum” error, should be replaced by aluminium;
Text has been changed. (page 2 line 8)
- Session “2. Materials and Methods” It would be interesting to let the reader know the relevant (mechanical and physical) properties and to have a comparison in terms of future work by other authors.
The mechanical properties of the composite wire obtained by this method and comparison with the properties of similar composites are described in [8].
Text has been changed. (page 3 lines 126-128)
- Figure 2, why is the study at 6, 6.6, 7, 8, 8.6 and 10 seconds? What is the logic? Why not study at 7.6 and 10.6 seconds, for example?
Since the fiber path in the cavitation region was 20 mm, integer speed values give fractional time values. So, for example, at speeds of 200 and 90 mm/min, it takes about 6 and 13.6 seconds to pass through the cavitation area, respectively.
Text has been changed. (page 4 lines 136-139)
- Figure 3: If the experimental results were obtained between 6 and 10 seconds (for example, figure 2), how do the authors present us with a trend line for values below 6 and above 10 seconds? And how do they justify that for values greater than 10 seconds, the trend is completely different (constant)? At least one value greater than 10 seconds must be added to the graph to justify this result. The same for values less than 6 values, to justify the trend.
Since the samples obtained at times 11.4 and 13.6 are similar to a fully infiltrated sample at 10 seconds, their structure is not shown here. In addition, it is logical to assume that a further increase in time will also ensure complete infiltration. Infiltration times less than 6 seconds were not investigated in this work.
Text has been changed. (page 4 lines150-153)
- Line 192: Figure 5 does not exist.
Figure 5 has been added. (page 8 line 206)
- The English must be revised and improved.
The English has been revised and improved. Changes can be seen in change tracking mode.
Reviewer 2 Report
Dear Authors:
I think after reading through the paper the introduction needs to be rearranged or modified. In the introduction, the equation 2 and 3 are same, which I believe is wrong. Why does the cavitation matter for Pressure rate and flow rate should be explained in introduction for clarification in a sentence or so.
The authors need to give extra care to the sentence completion. Line number 110 is many among them. The pictures in the result must be provided with colors for identification.
Where is Figure 5? Again correct placement of punctuations are needed in many places. I don't think in the current format we can accept this paper.
Thanks
Author Response
- I think after reading through the paper the introduction needs to be rearranged or modified. In the introduction, the equation 2 and 3 are same, which I believe is wrong.
Equation 2 has been change to correct form. (page 2 line 7)
- Why does the cavitation matter for Pressure rate and flow rate should be explained in introduction for clarification in a sentence or so.
At the moment of collapse of cavitation bubbles, microflows arise, the local pressure of which can reach several GPa. This results in an increase in pressure in the cavitation area.
Text has been changed. (page 3 lines 101-103)
- The authors need to give extra care to the sentence completion. Line number 110 is many among them.
Extra care to the sentence completion was given. Changes can be seen in change tracking mode.
- The pictures in the result must be provided with colors for identification.
The pictures in the result have been provide with colors. (page 6 line 166; page 7 line 190; page 8 line 206)
- Where is Figure 5?
Figure 5 has been added. (page 8 line 206)
- Again correct placement of punctuations are needed in many places.
Punctuation has been revised and improved. Changes can be seen in change tracking mode.
Round 2
Reviewer 1 Report
The improvements and corrections made by the authors have substantially increased the quality of the work. Excellent work.
Author Response
Thank you very much for your time!